# Sexuality in Kidney Transplant Recipients: A Qualitative Study

**DOI:** 10.3390/healthcare9111432

**Published:** 2021-10-24

**Authors:** Ximena Abarca-Durán, Isabel María Fernández-Medina, María del Mar Jiménez-Lasserrotte, Iria Dobarrio-Sanz, Ana Lucía Martínez-Abarca, Cayetano Fernández-Sola

**Affiliations:** 1Facultad de Ciencias Médicas, Universidad de Cuenca, Cuenca 010204, Ecuador; xilup@yahoo.com.mx; 2Department of Nursing, Physiotherapy and Medicine, University of Almeria, 04120 Almeria, Spain; mjl095@ual.es (M.d.M.J.-L.); ids135@ual.es (I.D.-S.); cfernan@ual.es (C.F.-S.); 3Facultad de Medicina, Universidad de las Américas, Quito 170124, Ecuador; ana.martinez.abarca@udla.edu.ec; 4Facultad de Ciencias de la Salud, Universidad Autónoma de Chile, Santiago 7500000, Chile

**Keywords:** kidney transplant, transplant recipients, sexuality, sexual functioning, qualitative study

## Abstract

End-stage kidney disease has a negative impact on patients’ quality of life. People who receive a kidney transplant experience an improvement in many areas of their daily life. Sexuality is a general component of health, which can be affected by end-stage kidney disease and kidney transplant. The aim of this study was to explore and understand the experiences and perspectives of kidney transplant recipients regarding their sexuality. A qualitative study based on Gadamer’s hermeneutic philosophy was carried out. Two focus groups and nine interviews were conducted with 18 kidney transplant recipients. Data were audio-recorded, transcribed, and analyzed with the help of qualitative analysis software. Two main themes emerged from the data: (1) “The impact of a kidney transplant on sexuality”, with the subthemes “sexuality is relegated to the background”, “physical decline acts as a sexual inhibitor”, and “changes in sexual activity following a kidney transplant”; (2) “Sexual education in kidney transplant recipients” with the subthemes “sexuality: a hidden concern amongst kidney transplant recipients” and “talking about sexuality with healthcare professionals”. Sexuality is a frequent concern among kidney transplant recipients. The physiological and emotional changes experienced after kidney transplant exert a great influence on their sexuality. Healthcare professionals rarely discuss sexuality concerns with kidney transplant recipients. Professional sexual education and assistance are necessary to improve sexual health satisfaction of kidney transplant recipients.

## 1. Introduction

End-stage kidney disease (ESKD), defined as the kidney’s inability to maintain electrolytes and fluid balances and cleanse the body of toxins [1], significantly reduces the quality of life of people who suffer from it [2]. It is estimated that around two million people worldwide are treated for ESKD [3]. The treatment of choice for ESKD patients and the only one capable of correcting it is kidney transplant [3], which substantially improves survival rates and wellbeing for ESKD patients compared to other replacement therapies [3,4]. Nevertheless, a kidney transplant is a notably stressful event, which involves numerous physical, psychological, affective, and relational changes both for the kidney transplant recipients and for their family [5].

Sexuality is considered an essential element of general health and a basic human right [6,7]. Sexuality can be described as a central aspect of being human throughout one’s life, which includes sex, gender identities and roles, sexual orientation, pleasure, eroticism, intimacy, and reproduction [7]. Sexuality is influenced by the interaction of several factors, among which the biological, psychological, and social factors stand out [7].

Sexual dysfunction is a highly prevalent problem in women and men with ESKD [8,9]. The prevalence in women is approximately 74% [10], while in men it is 82% [11]. The most common sexual dysfunctions are erectile dysfunction in men [11,12] and decreased vaginal lubrication, pain during intercourse, and difficulty reaching orgasm in women [10,13]. In addition, both men and women often experience a decrease in libido [14]. The etiology of sexual dysfunctions in ESKD patients is multifarious and includes comorbid conditions, treatment-related factors, and physiological and psychological states [8,15]. Nevertheless, others factors such as older age, diabetes mellitus, cardiovascular diseases, and depression increase the risk of sexual dysfunction in ESKD patients [15,16].

Some studies have shown that kidney transplants can have a positive impact on the sexuality of both men and women [6,7]. This positive impact may be due to the correction of metabolic and endocrine imbalances, the normalization of hormonal profiles, and the improvement of the psychological state [8,17]. Although most studies are conclusive regarding improvements in female sexuality after kidney transplant [17,18], the results of studies conducted in male kidney recipients are controversial [11,19]. After a kidney transplant, 30–50% of men reported erectile dysfunction or alterations in ejaculatory function [20,21]. In addition to comorbidities, the changes in the patient’s body image due to the surgery and immunosuppressive therapy may alter their sexual response [19]. The persistence of sexual dysfunctions after a kidney transplant has a negative impact on kidney transplant recipients’ wellbeing and life satisfaction [22]. However, in spite of a high incidence of sexual dysfunctions in ESKD and kidney transplant recipients, the issue is often undervalued and ignored by healthcare professionals in their daily practice [23].

Previous research on sexuality following a kidney transplant have been focused on sexual dysfunctions such as anorgasmia or erectile dysfunction [17,18,19], but few studies have explored the kidney transplant recipients’ experiences and perspectives about sexuality [24]. Therefore, the aim of this study was to explore and understand the experiences and perspectives of kidney transplant recipients regarding their sexuality.

## 2. Materials and Methods

### 2.1. Study Design

A qualitative study was designed according to Gadamer’s hermeneutic phenomenology. For Gadamer, the human experience can only be understood through language. The understanding of a phenomenon is achieved through a fusion of horizons between researchers (precompression) and participants (according to their experiences) [25]. The development of this study followed the phases of Gadamerian-based research [26,27]: 1. To decide whether the study question is relevant to the methodological assumptions. Sexuality in kidney transplant recipients is a phenomenon that can be explored from the perspective of hermeneutic phenomenology. 2. To identify the researchers’ previous understanding of the topic. The researchers’ previous understanding came from their clinical experience as healthcare professional having worked in transplant rooms or with patients with chronic kidney disease. This study followed the Consolidated Criteria for Reporting Qualitative Research (COREQ) [28].

### 2.2. Participants

Participants were recruited using a convenience sample technique amongst patients who had received a kidney transplant during the last five years in a hospital located in Quito (Ecuador). The first author invited patients to participate in the study via phone call. Patients were informed of the aim of the study and were asked if they were interested in participating in the study. Patients were involved in this study if they were ≥18 years old, had received a kidney transplant more than three months before, had a functional transplant, and had an active sexual life. Patients were excluded if they had received a kidney transplant less than three months before, did not have an active sexual life, or refused to participate in the study.

### 2.3. Data Collection

We conducted two focus group discussions with 9 participants and 9 in-depth interviews. The focus groups, with an average duration of 75 min, and the interviews, with an approximate duration of 40 min each, were carried out in a private office in the local university’s premises. To facilitate the emergence of relevant themes, which were later explored in the interviews, the focus groups were conducted first. The focus groups and interviews were conducted by X.A.D. with the help of A.L.M.A, as an observer. The interviewer had training in medicine (MD), a master’s degree in sexology, and is a PhD student. Her extensive experience covered the clinical and management field, as Vice Minister first, and Minister of Public Health of the Republic of Ecuador later. At the time of the interviews, she was the coordinator of transplants in Ecuador. The position did not imply direct contact with people but carried out management and coordination tasks between donations and transplants. A.L.M.A. is MD, PhD. Focus groups began with a general question: What has your experience been regarding sexuality after kidney transplantation? Although interviews were distinguished by an open approach, we used a guide interview to conduct interviews, which included themes belonging to sexuality attitudes, practices, and experiences about sexuality after kidney transplantation (Table 1). Data collection was concluded after reaching data saturation, the point at which no new information emerged from the data. Both focus groups and interviews were audio-recorded and transcribed verbatim. When the audio offered doubts, fragments of transcription were shown to some participants so that they could confirm the content to clarify doubts.

The Ethics and Research Committee of the hospital Eugenio Espejo approved this study. All participants participated voluntarily, received written information about the aim of the study, and signed an informed consent form. Anonymity and confidentially of participants were guaranteed.

### 2.4. Data Analysis

Data were analyzed using the following steps [26,27]: 1. To achieve an understanding of the topic through dialogue with the participants. During the interviews new questions emerged such as “How does kidney transplantation affect sexuality?”. 2. In order to understand them, each transcript was analyzed line by line. Data analysis was carried out individually by three researchers. Meaningful and important phrases were selected as citations. Each citation was assigned a code which captured its meaning. After discussing the previous analysis, the codes were grouped into units of meaning, themes, and subthemes. This step was fulfilled with the help of the computer qualitative data analysis program ATLAS.ti 9. 3. To assure the rigor of the study, reliability came from the detailed description of all steps of the research and data collection. Credibility was demonstrated by representing all of the opinions of the participants in the study. To establish confirmability, the final list of themes and subthemes was returned to the participants to confirm the results.

## 3. Results

A total of 21 kidney transplant recipients were invited to participate, but three refused to take part. The final sample was made up of 18 kidney transplant recipients (10 women and eight men) with a median age of 39 and an average of 17.7 months after kidney transplantation. Socio-demographic characteristics of the participants are shown in Table 2.

Emerging results were grouped into units of meaning, sub-themes, and themes (Table 3). All of them allow us to know and understand the kidney transplant recipients’ experiences of sexuality.

### 3.1. Theme 1: The Impact of a Kidney Transplant on Sexuality

A large proportion of ESKD patients are found to have some type of sexual problem. Sexual function is a frequent concern among men and women who have received a kidney transplant. This theme reflects how their sexuality was affected by their health concerns as well as exploring participants’ sexual experiences following the kidney transplant.

#### 3.1.1. Subtheme 1. Sexuality Is Relegated to the Background

Sexuality, which to most of our participants is related to coital relations, was a secondary concern as transplanted people, especially if only a short time had passed since the transplant took place. The kidney transplant recipients perceived that their priority was to survive and recover from the surgery performed.


*“For me right now sexuality is not essential, my sexual life is in a secondary place, I think about being healthy…I want to have a child but in this situation of being recently transplanted, first I have to make sure that I am well to give life….”*
(Interview 3)

Nevertheless, some patients said that both their partner and they delayed the initiation of sexual relations after the transplant because they were afraid of damaging the transplanted kidney. Most of our participants began to show interest in sexual intercourse and reproduction within the first year after surgery.


*“I was very afraid of having sex too soon after the transplant because I might damage the kidney in some way…. Although I felt pressured by my husband, I avoided having sex for at least the first year after the transplant…”*
(Interview 9)

Recently, kidney transplant recipients reported that they had experienced lack of desire and libido to participate in sexual activities, which they attributed to the renal functioning and chronic fatigue caused by the surgery. Because of this, some participants had to make a change from the sexuality model centered on intercourse towards a more comprehensive and affective sexual model. Although they accepted it with resignation, our participants felt guilty and worried about the impact of sexuality on their relationships with their partners. They considered that the support and understanding of their partner was essential at this stage. Most of the men who had received a kidney transplant received support from their female partners, which was not always the case for female kidney transplant recipients, many of whom were involved in situations of emotional breakdown.


*“Sexuality is a very important factor for a couple…but when you are sick with kidney failure and have received a kidney transplant, sex is not everything in a couple, there must be love and above all understanding…”*
(FG1)

#### 3.1.2. Subtheme 2. Physical Decline Acts as a Sexual Inhibitor

Chronic kidney disease, transplant surgery, and the use of immunosuppressant medications provokes significant changes in physical appearance and body image such as weight gain, hair growth, or scars. The alteration of body image caused our participants to not recognize themselves in the mirror and to feel less attractive. Some of them felt ashamed of their body and were afraid of being rejected by their partner, which led them to avoid having sex. This feeling was especially pronounced in women.


*“My main problem is my appearance… I have gained a lot of weight and my body is full of scars…it is difficult for my partner to see me naked and have sex….”*
(Interview 7)

Those participants who did not have a stable partner admitted that the alteration of their body image affected their self-esteem and they had difficulties in starting new relationships. The fear of being rejected for their physical appearance or not being able to have a satisfactory sexual relationship were the main factors that led our kidney transplant recipients to reject the start of new romantic relationships.


*“Nothing is the same, I have changed in such a way that I am no longer me… also in sexual relations it lasted longer before, now the duration is half of what it was before… do you think it’s possible that I will find a partner who accepts me like this? ”*
(Participant 9)

In addition, many of the kidney transplant recipients are usually elderly people who suffer from other diseases associated with sexual problems such as diabetes or heart diseases. These participants were pessimistic about their ability to have a sexual relationship after kidney transplantation.


*“I have been on dialysis for ten years, I also have diabetes, cholesterol… I lost vaginal lubrication due to kidney failure and diabetes, I felt a lot of pain; dissatisfaction with sexual relations leads you to avoid them….I don’t think that it will improve now following the transplant…”*
(FG1)

#### 3.1.3. Subtheme 3. Changes in Sexual Activity Following a Kidney Transplant

Following kidney transplantation, both men and women may notice changes in their sexual functioning. These sexual changes in our participants varied in intensity and occurred at different times. According to the opinion of our participants, the improvement in mood after kidney transplantation had a positive influence on their sexuality.


*“When I got sick, I didn’t want to think about myself…my mood changed a lot after the transplant, I had more desire to live, to be with my husband…and I began to be interested in sexuality, which I had not done before…”*
(FG1)

Although some of the women who had received a kidney transplant reported having experienced an improvement in reaching orgasm and less pain during sexual intercourse, others did not note any change in their sexual activity. Women associated increased vaginal lubrication with positive changes in their sexuality. In contrast, women who did not experience changes in their sexuality conceived the sexual act as a marital obligation.


*“Before the transplant I was a very cold person when it came to sex, I had no desire, sometimes I didn’t feel anything, sometimes something… but after the transplant I had desire and pleasure again…”*
(Participant 1)

The male kidney transplant recipients commented that their sexuality had deteriorated as the kidney disease progressed. However, after transplant, they felt more sexually active and they managed to maintain an erection for longer. Other participants reported that despite having experienced an improvement in erection, they had difficulty reaching orgasm, and once achieved, the amount of semen expelled was minimal. However, other participants, especially older men or those with blood pressure problems, found no positive effects of the transplant on their sexuality.


*“After the transplant everything has returned to activity, my sexual appetite has returned again and the hardness of my penis in sexual relations, if I complete the sexual act, it’s thanks to the transplant…”*
(FG2)

Regarding masturbation, most of the female kidney transplant recipients acknowledged not practicing it. However, in male kidney transplant recipients, it was a habitual practice. According to the participating men, masturbation was not practiced for pleasure, but because it was one of the medical recommendations to avoid genital infections.


*“Masturbation was recommended by an urologist, she told me that I had to masturbate once a day because a man needs to ejaculate, semen can accumulate and cause infections…”*
(Participant 6)

### 3.2. Theme 2. Sexual Education in Kidney Transplant Recipients

Information regarding sexuality received by kidney transplant recipients is often scarce. Most kidney transplant recipients have doubts about their sexuality that they do not know which healthcare professionals they need to consult. This theme shows kidney transplant recipients’ perceptions and experiences regarding their doubts concerning sexuality and sexual education received.

#### 3.2.1. Subtheme 1. Sexuality: A Hidden Concern amongst Kidney Transplant Recipients

Sexuality is a difficult subject to address with kidney transplant recipients due to the various beliefs and taboos that exist about sex. Kidney transplant recipients often talk openly about their disease or their quality of life before and after transplant, but not about their sexuality. Our participants suggested that religion, together with the absence of sexual education received at home, exerted a great influence on the open communication of their sexual concerns.


*“Before, talking about sexuality was very restricted… my parents were very religious and considered that talking about sexuality to their children was perverting them…. We did not have the sexual education that can now be found in schools or sexual information you can find on the internet, we learned about sexuality alone and it’s more difficult for us to talk about it…”*
(Participant 2)

Sexuality was conceived by male kidney transplant recipients as something intimate, unamenable, and censored that cannot be shared with anyone. However, some of the participating women affirmed sharing their sexual concerns with their partner or close relatives. Nevertheless, most of the participants, despite experiencing concerns and problems related to the sexual domain, did not seek professional help, since they considered that their sexuality was a private matter that had no solution.


*“My sexuality is a subject that nobody cares about. Before the transplant I already had problems with sex, I don’t think my sexual problems will be fixed if I go to a healthcare professional to tell them about my sexual difficulties…. How will they help me? There is nothing to do, it’s an issue that I have to control myself”*
(FG2)

#### 3.2.2. Subtheme 2. Talking with Healthcare Professionals about Sexuality

Open communication between the healthcare professional and the kidney transplant recipients is essential to be able to treat different sexual problems that may arise. The participants of our study reported that, although they rarely express their sexual concerns spontaneously, healthcare professionals also have difficulties in recognizing the sexual aspect of kidney transplant recipients. The sexual information that healthcare professionals gave to our kidney transplant recipients was scarce and was limited to physiological aspects such as hormonal regulation and onset of menstruation or pregnancy. However, our participants demanded more information about the potential effects of a kidney transplant and medications on their sexuality.


*“He (healthcare professional) only told me that I would have my period again and that if everything went well I could get pregnant later… but not about the effects of the kidney transplant or medication on sexual desire for example…“*
(FG2)

Some participants of our study reported that healthcare professionals should receive additional sexual training in order to properly discuss sexuality concerns with patients. Furthermore, they suggested that sexual information should be directed to both the kidney transplant recipient and his or her partner in order to avoid conflict in the relationship.


*“I think it would be good for my partner if the doctor were to call us as a couple to talk about our sexual situation, he (the doctor) knows how I am and what I can and cannot do, so that my partner understands that I am feeling down when it comes to sexuality”*
(Participant 8)

## 4. Discussion

The aim of this study was to explore and understand the experiences and perspectives of kidney transplant recipients regarding their sexuality. After a kidney transplant, recipients experience several sexual concerns [22,24]. The results of our study show that sexuality is not a priority for kidney transplant recipients. In contrast, other studies have emphasized that having a kidney transplant does not make sexuality less significant to patients [29]. This could be because most kidney transplants are performed in adults where sexuality is of great importance [30]. However, the fear of rejecting the organ or the uncertainty about the state of health after the transplant are stressors that could influence the sexual interest of kidney transplant recipients [31]. Therefore, as our study reports, sexual interest decreased significantly in the posttransplant period, and most recipients postponed sexual intercourse [32]. According to the study carried out by Martell et al. (2015), fatigue is one of the factors that contributes to kidney transplant recipients’ lack of desire to participate in sexual activities. During the early posttransplant stage, kidney transplant recipients demonstrate greater interest in affection and the emotional bond in the relationship than physical sex [24]. Kidney transplant not only affects the recipient’s sexuality, but also that of their partner [8]. Lack of desire and changes in sexual functioning can lead to poorer relationship functioning [33]. In this way, the presence of a supportive partner is essential to cope with changes to sexuality posttransplant [24,34]. Supportive partners are associated with fewer negative sexual concerns, whilst the lack of a supportive partner contributes to increasing the recipient’s sexual concerns and frustration following their kidney transplant [24]. In contrast, Ragi et al. (2014) affirmed that, although both sexual desire and the frequency of sexual intercourse decreased among kidney transplant recipients and their partners, in the context of a relationship, there was no impact on their sexual satisfaction. This could be explained because sexual satisfaction is a multidimensional construct, which is not only correlated with sexual function but also with psychological and relationship aspects [35]. In addition, other types of sexual activities, for example, kissing, touching, or cuddling, are also essential elements in sexual satisfaction, especially in long-term relationships [35,36].

Kidney transplant recipients go through a prolonged period of adaptation to disease, which involves several body changes [37]. The use of kidney replacement therapies such as dialysis before the transplant provokes important changes in body image due to uremia effects and the creation of the dialytic access [37,38]. Furthermore, the integration and adaptation of a new kidney into the body implies an adjustment of body image, which can influence intimacy and sexual behaviors [39]. The study of Mota et al. (2019) corroborated the strong impact of body image satisfaction on sexual function in kidney transplant recipients. As per our results, the changes in physical appearance affected sexuality negatively [19,24]. Negative body image satisfaction is associated with higher levels of anxiety and depression in kidney transplant recipients [19]. Anxiety and depression are psychological states that are related to a decrease in sexual desire and functioning [14,40,41]. It could justify those participants of our study who experienced a positive change in mood having more sexual desire. Despite some of our participants experiencing an improvement in sexual functioning, sexual dysfunction is a frequent sexual problem among kidney transplant recipients [21,22]. Kurtulus et al. (2017) affirmed that age is the most important risk factor for sexual dysfunction. However, other factors in kidney transplant recipients have demonstrated a positive correlation with sexual dysfunction such as vascular comorbidities, the time undergoing dialysis, and medications [19,42].

Given that sexuality is an import part of kidney transplant recipients’ quality of life [29], it is important that healthcare professionals communicate information to recipients and their partners about the possible effects of kidney transplant and treatment on sexual wellbeing in order to normalize sexuality in this context. Furthermore, sexual adjustments made by kidney transplant recipients should be guided by healthcare professionals in order to promote sexual relations [43]. Most of our participants did not seek professional help for their sexual problem, as other studies have shown [24]. Van Ek et al. (2017) suggest that kidney transplant recipients do not usually seek help because sexuality is ignored by healthcare professionals during posttransplant consultation [44]. Healthcare professionals may not provide sufficient sexuality information to kidney transplant recipients for several reasons. On the one hand, and in line with our results, sexuality is still a taboo subject even today, which can cause discomfort in the patient and in the healthcare professional when discussing it [45]. On the other hand, healthcare professionals’ knowledge about sexuality may be limited [44,45]. For this reason, it would be important to include sexuality as another part of patient care in the training of healthcare professionals. Regardless, healthcare professionals do not consider sexuality a priority in the care of the patients, so they do not take this discussion into account.

## 5. Study Strengths and Limitations

Our results expand on those of prior qualitative studies referring to sexuality and kidney transplants. Participants’ characteristics are diverse in age and length of time following kidney transplant, which ensured the variability of the responses. However, this study is not without limitations. Firstly, the results of this study were influenced by the social and cultural factors of Latin American societies. In order to compare the results, similar studies could be conducted in other societies. Secondly, all participants were heterosexual so the results could not be extrapolated to other population groups with different sexual tendencies. Furthermore, this study only reflects the experiences and perspectives of kidney transplant recipients. Future studies should include the perspectives of kidney transplant recipients’ partners and healthcare professionals regarding sexuality after a kidney transplant in order to obtain a deeper understanding of the topic. To finish, the use of a convenience sampling technique may have limited the results.

## 6. Recommendations for Practice

This study contains relevant information for healthcare professionals, as sexuality is an important issue for kidney transplants recipients, and it is essential to include it in their care plan. Healthcare professionals should be aware that kidney transplant recipients can experience sexual concerns and problems. Providing kidney transplant recipients with clear information and by discussing sexuality and potential side effects of treatments would allow them to adopt new conceptions and sexual practices in order to maintain their sexuality, self-esteem, and quality of life. Consequently, it is necessary to train all healthcare professionals involved in the care of kidney transplant recipients in sexuality. Additionally, this study suggests that the partners of kidney transplant recipients are affected by their sexual changes. Therefore, sexual information and education given by healthcare professionals should be directed at both the kidney transplant recipient and their partner.

## 7. Conclusions

This study indicates that, although the main priority of kidney transplant recipients is to maintain the integrity and functioning of the transplant organ, sexuality is a frequent concern amongst them. Physiological and emotional changes after the kidney transplant affected sexual desire and functioning. Current support from healthcare professionals overlooked kidney transplant recipients’ sexual concerns and needs. Healthcare professionals need adequate training to be able to give adequate sexual advice and counselling to kidney transplant recipients and their partners. Sexual information and assistance could constitute an easy way to optimize the global sexual health satisfaction of kidney transplants’ recipients.

## Figures and Tables

**Table 1 healthcare-09-01432-t001:** Interview protocol.

Subject	Content and Example of Questions
My intention	I am a member of a research group that studies sexuality and kidney transplants. Knowing about your experiences could help to improve care in kidney transplant recipients.
Ethical issues	Participation is voluntary, you can abandon the study at any time you wish. The conversation will be recorded in order to analyze the data. Your personal data will not be disclosed.
Consent	Participants’ verbal acceptance and signing of the informed consent.
Introductory question	What is the importance of sexuality in your life?
Conversation guide	What is the importance of sexuality in your life? Could you tell me how you feel when you look in the mirror? Could you tell me how your sex life has changed since the kidney transplant? What are the main differences that you have perceived in your sexual relations following the transplant? What have been the main sexual concerns that have arisen since the transplant? What kind of sex education have you received since the transplant?
Final question	Would you like to add something else?
Appreciation	Thank you for taking part. Once the study has been completed, you will receive a copy.

**Table 2 healthcare-09-01432-t002:** Socio-demographic characteristics of the participants.

Socio-Demographic Characteristics	Focus Groups (*n* = 9)	Interviews (*n* = 9)
Female	5	5
Male	4	4
<30	2 (22.2%)	1 (11.1%)
31–40	3 (33.3%)	3 (33.3%)
>41	4 (44.4%)	5 (55.5%)
Single	3 (33.3%)	1 (11.1%)
Married	5 (55.5%)	5 (55.5%)
Divorced	1 (11.1%)	3 (33.3%)
3–14	2 (22.2%)	3 (33.3%)
15–24	4 (44.4%)	6 (66.6%)
>24	3 (33.3%)	0

**Table 3 healthcare-09-01432-t003:** Themes, subthemes, and units of meaning.

Theme	Subtheme	Units of Meaning
**1. The impact of a kidney transplant on sexuality**	1.1. Sexuality is relegated to the background.	Fear Lack of desire New sexuality models
	1.2. Physical decline acts as a sexual inhibitor.	Body image Difficulty in sexual relations Influence of comorbidities
	1.3. Changes in sexual activity following a kidney transplant.	Influence of mood Improvements in sexuality Masturbation
**2. Sexual education in kidney transplant recipients**	2.1. Sexuality: a hidden concern amongst kidney transplant recipients.	Taboo subject Denial Lack of sexual education
	2.2. Talking about sexuality with healthcare professionals.	Lack of information Lack of sexual training amongst healthcare professionals Partners

## Data Availability

For confidentiality purposes, the data are in the possession of the main author (X.A.-D.). The transcripts of the interviews and the focus groups are in a hermeneutic unit of ATLAS.ti, and this software is required to access them.

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
