# Peer review of "Sexuality in Kidney Transplant Recipients: A Qualitative Study"

_healthcare, 2021, doi:10.3390/healthcare9111432_

Round 1

Reviewer 1 Report

Dear Authors

Thank you for the opportunity to review your article.

Brief summary: This is a qualitative study based on Gadamer hermeneutic philosophy that aimed to explore and understand the experiences and perspectives of kidney transplant recipients regarding their sexuality.

Areas of strength

The references included are relevant for the subject under study show 26/45 (58%) references from the last 5 years. There is strong concordance between the and the methods used. The results are clearly described. The discussion correlates with the presented data and takes the published literature into account. The manuscript presents some limitations and practical implications.

Weakness:

  1. The description of the methodology wasn’t made clearly and adequately in some topics, as the authors reported following the consolidated criteria for reporting qualitative research (COREQ) of the

The authors should aim to respond objectively to the following COREQ criteria to improve the quality of the manuscript:

Domain 1: Research team and reflexivity

  1. Personal characteristics Interviewer/facilitator - Which author/s conducted the interview or focus group?
  2. Credentials - What were the researcher’s credentials? E.g. PhD, MD
  3. Occupation - What was their occupation at the time of the study?
  4. Gender - Was the researcher male or female?
  5. Experience and training - What experience or training did the researcher have?
  6. Review the referencing style. Put the year in bold and the journal abbreviations.

Author Response

Thank you for your comments. 

  • The interviews and focus groups were conducted by XAD (with the help, as an observer, of ALMA)
  • Credentials: XAD is MD, sexology MSc, PhD student. ALMA is MD, PhD.
  • Occupation: XAD was the transplant coordinator for Ecuador at the time of the study. The position did not imply direct contact with people, but carried out management and coordination tasks between donations-transplants.
  • Gender: Both interviewer and observer are women.
  • Experience and training: The interviewer had a medical background (MD), a master's degree in sexology, and a PhD student. Her extensive experience covered the clinical and management field, as Vice Minister, first, and Minister of Public Health of the Republic of Ecuador later. At the time of the interviews, she was the coordinator of Transplants in Ecuador. 

Reviewer 2 Report

Abarca-Durán X, et al. presented a qualitative study about sexuality in kidney transplant recipients in Quito, Ecuador. They used a focus group technique with a guide interview to qualitatively assess the participants' sexuality after kidney transplantation. This is an interesting study with intriguing results. I have a few comments. 

  1. Since the authors used the convenience sample technique, would the authors comment about this issue in the limitations?
  2. Would the authors provide the personal characteristics of 2 interviewers into the manuscript?
  3. Have the participants provided the correction and/or feedback on the transcription and findings? Please add this into the manuscript.

Author Response

Thank you for your comments. 

  • Convenience sampling has been mentioned in the limitations.
  • The characteristics of the authors have been added.
  • Comments about the audio and transcript have also been added. 

Round 2

Reviewer 1 Report

Dear authors

Thanks for the opportunity to review the article again. Congratulations, the article has been improved and its quality increased. 

The checklist Consolidated criteria for Reporting Qualitative research (COREQ) was followed in full.